# Effect of Silymarin Supplementation in Lung and Liver Histological Modifications during Exercise Training in a Rodent Model

**DOI:** 10.3390/jfmk6030072

**Published:** 2021-08-30

**Authors:** Nancy Vargas-Mendoza, Marcelo Angeles-Valencia, Ángel Morales-González, Mauricio Morales-Martínez, Eduardo Madrigal-Bujaidar, Isela Álvarez-González, Tomás Fregoso-Aguilar, Luis Delgado-Olivares, Eduardo Osiris Madrigal-Santillán, José A. Morales-González

**Affiliations:** 1Laboratorio de Medicina de Conservación, Escuela Superior de Medicina, Instituto Politécnico Nacional, Plan de San Luis y Díaz Mirón, Col. Casco de Santo Tomás, Del. Miguel Hidalgo, Ciudad de México 11340, Mexico; nvargas_mendoza@hotmail.com (N.V.-M.); angeles_v_marcelo@hotmail.com (M.A.-V.); 2Escuela Superior de Cómputo, Instituto Politécnico Nacional, Av. Juan de Dios Bátiz s/n Esquina Miguel Othón de Mendizabal, Unidad Profesional Adolfo López Mateos, Ciudad de México 07738, Mexico; anmorales@ipn.mx; 3Licenciatura en Nutrición, Universidad Intercontinental, Insurgentes Sur 4303, Santa Úrsula Xitla, Alcaldía Tlalpan, Ciudad de México 14420, Mexico; mtz98mauxd@gmail.com; 4Laboratorio de Genética, Escuela Nacional de Ciencias Biológicas, Instituto Politécnico Nacional, Unidad Profesional A. López Mateos, Av. Wilfrido Massieu, Col., Lindavista, Ciudad de México 07738, Mexico; eduardo.madrigal@lycos.com (E.M.-B.); isela.alvarez@gmail.com (I.Á.-G.); 5Laboratorio de Hormonas y Conducta, Departamento de Fisiología, ENCB Campus Zacatenco, Instituto Politécnico Nacional, Ciudad de México 07700, Mexico; tfregoso@ipn.mx; 6Centro de Investigación Interdisciplinario, Área Académica de Nutrición, Instituto de Ciencias de la Salud, Universidad Autónoma del Estado de Hidalgo, Circuito Actopan-Tilcuauttla, s/n, Ex Hacienda La Concepción, San Agustín Tlaxiaca 42160, Hidalgo, Mexico; ldelgado@uaeh.edu.mx

**Keywords:** exercise training, silymarin, lung, liver, histological changes, inflammation

## Abstract

Background: Exercise training induces adaptive physiological and morphological modifications in the entire organism; however, excessive loads of training may increase damage in tissues. The purpose of this study was to evaluate the effect of silymarin in lung and liver histological changes in rats subjected to exercise training (ET). Methods: Male Wistar rats were subjected to an 8-week ET treadmill program 5 days per week, 60 min/session, and were previously administered 100 mg ascorbic acid or 100 mg of silymarin. Results: Silymarin increased alveolar and bronchial muscle size, improve vascularization, and reduced tissue inflammation. In liver, silymarin promoted the reduction of lipid content. Conclusion: Silymarin supplementation may improve inflammation in pulmonary tissue after 8 weeks of the ET treadmill program, improve cell recovery, and reduce intrahepatic lipid content.

## 1. Introduction

A great amount of evidence supports the idea that exercise in different forms improves health and well-being, such as in the prevention of diseases or as a therapeutic option. The practice of physical exercise is responsible for inducing a variety of modifications in the whole organism as a result of adaptation. Normally, the body is able to adjust to physical work, but when physical demands are greater than the body’s ability to compensate excessive workloads, imminent damage will emerge, affecting physical performance depending on the type, intensity, volume, and frequency of the exercise [1,2]. Hence, it is important to give the body the opportunity to recover from the wasting induced by exercise training (ET). The capacity of the organism to cope with high demands depends on the equilibrium between the level of the exercise-induced damage and the endogenous and exogenous defense systems [3].

Silymarin, the name given to the extract of the species derived from *Silybum marianum*, has been traditionally used for many years in ancient medicine in the therapy of liver injuries [4] and it has been recognized as possessing exceptional antioxidant properties. The use of silymarin in physical exercise protocols is limited and very few studies have been carried out to prove its effectiveness in enhancing physical capacity during exercise. The elevated demand of oxygen (O_2_), fundamentally in endurance ET, promotes the inflammatory response along with modifications of the airways, and is accompanied by changes in other organs such as muscle and myocardium. It appears that the use of silymarin exhibits a benefit on the recovery and hypertrophy of quadriceps and gastrocnemius muscle and the myocardium, together with an increase in maximal endurance capacity and the improvement of the lean muscle mass and body weight of rats subjected to regular exercise training, as our research has previously reported [5]. During exertion, there is a multi-organ-function coupling triggered by muscle power in a demand for O_2_-rich blood; thus, pulmonary function is basically implicated in O_2_ and CO_2_ exchange and transport. However, primarily in aerobic exercise, the mechanisms involved in the control of ventilation are related to the function of the respiratory muscle, pulmonary vascular function in crosstalk with cardiac output, and the requirement of blood flow in active muscles [6]. In the liver, the changes could be associated with the glycogen content in response to glucose metabolism and to the lipid content observed fundamentally in prolonged endurance exercise [7]. The role of the liver is crucial to meeting metabolic demands in working muscles; this organ is responsible for storage and release of energy substrates in the form of glucose or fatty acids. In this regard, the liver can adapt in the same manner as the muscles to regular exercise by increasing its ability to oxidize fat [8].

Indeed, exercise dynamics promote certain modifications in organs and systems that can be observed by mean of the analysis of tissues. It is noteworthy that, at a certain volume, intensity, or frequency, physical work induces stress. The organism can manage stress through adaptations. However, excessive workloads may impair the ability to deal with this stressful environment; thus, in turn, causing a delay in recovery or in the reduction of physical performance. Inflammation is one of those response mechanisms to physical stress that can be reflected through different indicators. For instance, histological modifications in the lung show the important role that lung parenchyma plays in gas exchange by increasing ventilation and oxygen consumption; during this process, the change in the respiratory pattern is observed, frequently from nasal to oral, which allows the entry of air with a higher proportion of polluting particles inducing the immune system response such as activation of pulmonary macrophages and cellular infiltrate [9,10]. During high intensity or strenuous exercise splanchnic organs such as the gut, liver, and pancreas suffer a blood flow reduction of up to 80% which may compromise physical performance [11]. In the liver the response to physical stress could be associated with activation of Kupffer cells and the inflammatory reaction appearance as well as a reduction in the lipid content of hepatocytes because of energy demands [12].

In this study, we evaluated the effect of silymarin supplementation on histological changes in lung and liver tissue after an 8-week treadmill ET program in a rodent model to make a proposal for the reasonable use of such phytochemical that may aid in the improvement of tissue recovery during high workloads. The previously mentioned effect could be observed by the assessment of specific structural modifications in tissues. This idea is supported by the previously reported findings.

## 2. Materials and Methods

### 2.1. Experimental Design

Male Wistar rats with a body weight of 200–200 g and 8-weeks-of-age were placed in environmentally controlled facilities. The animals were adapted to a room temperature of 22 °C with a 12 h:12 h light/dark cycle 1 week prior to the beginning of the experiment. The rats were fed with Chow formula 5008 (PMI Nutrition International, LLC., Concord, NH, USA), which contained 23% protein, 6.5% fat, 4% fiber, 12% humidity, and 8% ash, and water ad libitum. Animals were randomly divided into the following four groups (*n* = 5): control group, exercise training group (ET), exercise training and ascorbic acid (ET+Aa) group, and exercise training plus silymarin (ET+Sm) group. The ET+Aa and ET+Sm groups were administered intragastrically (i.g.) with 100 mg/kg of ascorbic acid and 100 mg/kg of silymarin, respectively, 30–40 min prior to the training session. Ascorbic acid was considered as a control because of its the impact on recovery and the immune system, and in addition, antioxidant activity was reported in some exercise training models in both animals and humans [13,14]. All procedures were performed according to the Official Mexican Guidelines for Laboratory Animal Use and Care (NOM-062-ZOO-1999) and were approved by the Comité Interno de Bioética of the Instituto de Ciencias de la Salud, Universidad Autónoma del Estado de Hidalgo, Mexico, approval number CIECUAL/012/2019.

### 2.2. Exercise-Training (ET) Protocol

Animals were subjected to adaptation during the physical-training phase on the treadmill (Motor Treadmill Pro-for 305 CST adapted for rodent) during 1 week. On day 1, the animals ran at a speed of 16.6 m/min, 0 degrees of inclination for 15 min; on day 2, animals ran 16 m/min, 0 degrees for 15 min followed by 18.33 m/min, 5 degrees of inclination for 15 min more, and on day 3, animals ran 16 m/min, 0 degrees for 10 min, followed by 18.33 m/min, 5 degrees for 10 min and ending at 20 m/min, 10 degrees for 10 min. Days 4 and 5 were the same as day 3. After the adaptation phase, the animals trained during the complete session running at 16 m/min, 0 degrees, for 10 min, followed by 18.33 m/min, 5 degrees for 10 min, and ending at 20 m/min, 10 degrees until completing the 60-min training session, 5 days/week for 8 weeks. The described ET protocol is based on a previously described training model [15].

### 2.3. Sample Collection

After the last training session in week 8, the rats were anesthetized with phenobarbital sodium (Pisa, Mexico City, Mexico) (40 mg/kg) and sacrificed by cervical dislocation. Lungs and liver tissues were obtained and embedded in formaldehyde (Merk, Kenilworth, NJ, USA) solution at 10% to preserve them. Tissues were then stained with hematoxylin and eosin (H&E).

### 2.4. Microscopy Analyses

Samples of lung and liver were processed for H&E staining according to the previously described technique [16]. Samples were cut, fixed, and paraffin-embedded. Glass slides holding paraffin (Leica Biosystems, Nussloch, Germany) fragments were placed in staining racks. The samples were cleaned of paraffin during three changes of xylene for 2 min per change, and then hydrated; afterward, the slides were transferred into three changes of ethanol (Merck, Kenilworth, NJ, USA) (100%) for 2 min/change; samples were transferred through ethanol (95%) for 2 min and then for 2 min more in ethanol (75%). Samples were cleaned in running tap water at room temperature for 2 min. Samples were stained with the hematoxylin solution (Merck, Kenilworth, NJ, USA) for 3 min and washed under running tap water at room temperature for 5 min, followed by staining with eosin Y (Certistain^®^, Merck, Kenilworth, NJ, USA) solution for 2 min and then were dehydrated by dipping the slides into ethanol (95%) 20 times. The slides were transferred for 2 min into ethanol (95%) and transferred once again during two changes of ethanol (100%) at 2 min/change. Finally, the samples were cleaned in three changes of xylene (Sigma-Aldrich, Merk KGaA, Kenilworth, NJ, USA), 2 min/change, and one drop of Permount was placed above the tissue on each slide with a coverslip (Merck Millipore, Kenilworth, NJ, USA). Lung and liver samples were observed using electronic microscopy (Olympus CH30). Tissue analyses were performed according to the criteria of Heffner et al. [17] for lung tissue considering the following: 0 = Null (no presence/appearance); + = Low (minimal presence/appearance); ++ = Moderate (intermediate presence/appearance); +++ = High (important presence/appearance), ++++ = Remarkably high (notable presence/appearance). The criteria of Morales-Gonzalez et al. [18] were employed for liver tissue considering the following: 0 = Null (no presence/appearance); + = Low (minimal presence/appearance); ++ = Moderate (intermediate presence/appearance); +++ = High (important presence/appearance), and ++++ = Remarkably high (notable presence/appearance). Each tissue was analyzed for specific characteristics. Evaluation of the lung tissues included the following: (1) alveolar size; (2) type 1 pneumocytes (T1P); (3) type 2 pneumocytes (T2P); (4) vascularization; (5) inflammation; (6) bronchial muscle hypertrophy; (7) connective tissue; (8) alveolar wall thickness, and (9) thickness of wall tissue. Evaluation of the liver tissues were as follows: (1) inflammatory reaction; (2) Kupffer cells; (3) steatosis; (4) mitosis; (5) apoptosis/necrosis; (6) hyperchromatic nuclei, and (7) nuclear enlargement.

## 3. Results

### 3.1. Histological Analysis

#### 3.1.1. Lung Tissue

Histological modification in lung tissue after 8 weeks of ET with ascorbic acid and silymarin are presented in Table 1. It can be observed than the control group did not show significant changes in alveolar size, nor in the number of type 1 or type 2 pneumocytes; we observed medium vascularization, and null inflammation and bronchial muscle hypertrophy as long as connective tissue remained at the baseline level. On the other hand, the thickness of the alveolar wall maintained an intermediate size and pleural tissue was slight (Figure 1a). The ET and ET+Aa groups demonstrated increased alveolar size, while type 1 pneumocytes remained the same in all groups. Type 2 pneumocytes increased to moderate size in the ET+Aa and in ET+Sm groups compared to the low level reported in the ET group (Figure 1b). In addition, vascularization augmented notably in all trained groups, but the level of vascularization in the ET+Sm group was remarkable. The inflammatory reaction was high in the ET group, while in the ET+Aa group it was moderate and it remained low in the ET+Sm group, as in the control group. In parallel, a greater increase in size of bronchial smooth muscle was observed in the group administered with silymarin comparted to the ET and ET+Aa groups (Figure 1c,d). Instead, the connective tissue size increased in ET group; however, in the ET+Aa and the ET+Sm groups, this remained small. The pleural wall increased in width in the ET group, while in the ET+Aa it achieved medium size and the pleural wall appearance in ET+Sm was narrow. Finally, a moderate increase was observed in the thickness of pleural tissue in the ET group, while in the groups administered with ascorbic acid and silymarin, this thickness remained low.

#### 3.1.2. Liver Tissue

Histological changes in the liver tissue of animals subjected to an 8-week ET program supplemented with ascorbic acid and silymarin are presented in Table 2. The control group remained without inflammatory response; in addition, low vascularization and the moderate presence of Kupffer cells were observed. Instead, the control group exhibited the highest level of steatosis, in the absence of mitosis and apoptosis/necrosis; there were also no changes in nuclear size or in hyperchromatic nuclei (Figure 2a). In contrast, the ET group reported augmented vascularization and a reduced steatosis level. The inflammatory response and the presence of Kupffer cells remained the same as those of the control group, whereas the signs of mitosis and apoptosis/necrosis were null. Hyperchromatic nuclei and moderate nuclear enlargement were observed (Figure 2b). On the other hand, the ET+Aa group presented a low inflammatory response, while this response was null in the ET+Sm group. The presence of Kupffer cells was high in both groups. Importantly, the level of vascularization increased in the ET+Aa and ET+Sm groups compared with the control and ET groups, while the level of steatosis was minimized in the antioxidant-supplemented groups. No indicators of mitosis, hyperchromatic nuclei, or the nuclear enlargement of hepatocytes were found. The ET+Aa group presented low apoptosis/necrosis, while the ET + Sm group was null (Figure 2c,d).

## 4. Discussion

Extensive evidence supports the idea that the regular practice of physical exercise promotes well-being and health, contributing to weight loss, body fat reduction, and lean muscle mass gain. Regular exercise induces morphophysiological changes that permit the organism to adjust to mechanical workloads according to frequency and intensity. However, excessive training may lead to cell and tissue damage, impairing recovery and physical performance. In this regard, the main objective of this study was to evaluate the impact of silymarin on lung and liver tissue repair in rats subjected to an 8-week physical training program. Silymarin is the name given to a group of compounds extracted from *Silybum marianum* species, it is traditionally known as milk thistle or St Marie’s thistle and has been employed in the treatment of liver disorders from ancient times by diverse civilizations [19]. A wide range of evidence shows different biological activities such as being a cardio, renal, and lung protector, and a hepatoprotector [4,20,21]. These activities have been associated with the potent antioxidant effect described [22]. Silymarin extract is basically composed of a group of flavonolignans, with silybin being the major one and comprising around 50–70% of the extract, followed by the remaining constituents and their isomers [19]. Many clinical trials have been conducted to prove the effectiveness of silymarin or its flavonolignans in several medical disorders; however, information on the effects of physical exercise is very scarce. Previously, one study reported that the administration of silymarin (50 mg/kg) may improve the metabolism of glucose, lactate, and lipids during exercise by interfering with the expression of the genes involved in gluconeogenesis, β-oxidation, and glucose oxidation in a 4-week treadmill running protocol in rats [23].

After 8 weeks of treadmill ET, changes in lung tissue were observed in all trained groups. the ET and ET+Aa groups reported larger alveolar size, whereas in the ET+Sm group, the increase in size in the silymarin group was more noticeable than in the other trained groups, in conjunction with remarkably augmented vascularization. Higher hypertrophy of the bronchial muscle was also observed in the group treated with silymarin than in the ET and ET+Aa groups. During endurance exercise, a constant supply of O_2_ is required to ensure the viability of ATP within oxidative metabolism. This O_2_ is transported and incorporated from the air into the environment of the upper airways (nose, pharynx, and trachea) to the lower airways (tracheobronchial tree and lungs), where it binds to hemoglobin in the erythrocytes. It is then transported by the blood to the cells, while the CO_2_ produced by the metabolism emerges and is carried to the lungs in the blood for its removal [24].

In physical training, lungs receive 100% of the blood flow ejected by the heart; this implies an exercise-induced lung stress. With incremental physical exertion, the pulmonary diffusion capacity (PDC) must increase to meet the demands of O_2_ [25]. From rest to the maximal peak, the PDC can increase up to 150% of the values of rest during physical exertion, without reaching an upper limit with respect to cardiac output. PDC increases as a result of increased capillary blood volume (CBV) and membrane diffusion capacity (MDC), secondary to the recruitment and distention of the pulmonary capillaries [26].

In aerobic endurance sports, there is a higher consumption of O_2_, thus a greater need for diffusion, which leads to an optimized PDC due to the increase in MDC but not in CBV. This suggests an adaptation in the athlete’s lung membrane [27]. The increases in MDC and CBV during exercise are achieved due to the increase in pulmonary arterial pressure, which results in the recruitment and distention of pulmonary capillaries that have previously been hypoperfused during the resting state [27]. Such a situation favors the increase in the capillary network of the lung cross-sectional area (CSA), reducing vascular resistance and attenuating pulmonary arterial pressure. Some evidence has shown that during exercise, there may be a recruitment of intrapulmonary arteriovenous anastomoses, intervening in gas exchange or in reducing pulmonary arterial capacity [27,28,29]. All of these physiological changes in lung tissue also induce cellular morphological modifications. Probably, as a necessity to compensate for the increase in blood load and diffusion, this could have resulted in the occurrence of alveoli size modifications accompanied by vascularization and enlargement of bronchial muscle in all trained groups. Nonetheless, the effect was importantly noticeable in the group administered with silymarin. On the other hand, connective tissue in the antioxidant groups remained low. The thickness of the wall and pleural tissue increased significantly in the ET group, while it diminished in the antioxidant-supplemented groups, especially in the one treated with silymarin.

On the other hand, the population of type 1 pneumocytes was similar in all groups, while type 2 pneumocytes were moderate in the ET+Aa and ET+Sm groups, as in the control. The inflammatory phenomenon in the EF group was remarkably high, while in the EF+Aa group it was moderate, and in the ET+Sm group it decreased to control levels. Type 1 pneumocytes comprise around 95% of the alveolar surface and are responsible for carrying out gas exchange. They are thin cells and extend along the alveolar wall, and they are sensitive to toxic substances. While type 2 pneumocytes form the remaining 5% of the cellular alveolar structure, they are found in the alveolar septal junctions, and their main function is the synthesis of surfactant liquid that reduces the surface tension of the alveolus, allowing for gas exchange and preventing adhesion of the lateral alveoli. The response to exercise involves stress along with the inflammatory response; the degree and manifestation has to do with whether the training stimulus is acute or chronic. Previous studies in rodents have revealed that 6 weeks of physical training for 5 days/week at moderate intensity increases leukocyte infiltration in the respiratory tissue; additionally, the induction of apoptosis and the decrease of hair cells have been observed [30].

Some studies in humans indicate that high-performance athletes such as runners or swimmers have permanently elevated inflammation biomarkers without implying the clinical manifestation of a disease. The markers observed include elevated polymorphonuclear cells, eosinophils, interleukin-8 (IL-8), leukotriene E4, prostaglandins, and histamine, TNF-α, and low blood pH [31,32]. Chronic inflammation has been found associated with lung epithelial damage in swimmers during 20 weeks of pool training in chlorinated water [33]. In general, animal models tend to show a homogeneous increase in inflammation indicators, but in human studies, researchers have paid attention to adjacent factors such as the pollutants present in the environment where the athlete performs their training. For runners exposed to training bouts in the open air or at high altitudes, the temperature and humidity of the inspired air, the meters above sea level, the hypoxic conditions, or the pollutants dissolved in the inspired air can have a significant effect on inflammation markers. In swimmers, due to chlorinated substances in the water of the swimming pools or other compounds in the water of open places the permanent infiltration of granulocytes, macrophages, and lymphocytes has been found, as well as of other, already mentioned, inflammatory substances [34]. The aforementioned chronic inflammation is associated with changes in lung tissue and cell remodeling, which may be associated with functional changes and persistent symptoms [35]. In the present study, we observed that the 8-week ET induced the formation of type 2 pneumocytes, probably as a compensatory mechanism in response to inflammation and the increasing demand for gas exchange and diffusion, ensuring the appropriate production of surfactant liquid, while increasing the inflammatory response to training-induced stress. The effect of silymarin over remodeling in lung tissue is visible at different levels—alveolar size, bronchial muscle hypertrophy, alveolar wall thickness, and inflammation. Thus, it would be reasonable to assume that the described antioxidant activity of silymarin had contributed to creating a better environment for favoring recovery from stress induced by exercise. In addition, silymarin has been associated with protein synthesis by promoting DNA synthesis and stimulating RNA polymerase I enzymes may enhance rRNA transcription. This implies a greater availability of structural and functional proteins to improve cell conditions for recovery and growth [36,37].

Supplementation with antioxidants in sport has been an issue for a long time. On the one hand, a certain degree of oxidative stress is necessary to unleash cytoprotective and adaptive cellular responses. However, it has also been proven that this efficient cell response can be attenuated by the use of exogenous antioxidants. Therefore, beyond the ergogenic effect, antioxidants might not benefit physical performance [38]. However, professional athletes are often exposed to a highly demanding training and in such case it could be rationale to give an extra aid to cover these needs. Vitamin C or ascorbic acid is considered a nutrient that may contribute to attenuating oxidative stress and inflammation by quenching ROS and reducing interleukin-6- and cortisol-enhancing immune response to damage cause by physical training in athletes [14]. Likewise, the International Society of Sports Nutrition classifies vitamin C as a supplement I with strong evidence to prove the efficacy and apparent safety [13], but the results of studies are controversial; some of them support the ascorbic acid effectiveness exclusively in low plasma levels, that is in terms of deficiency [39]. The benefit reported from the use of vitamin C in physical training includes attenuation of bronchoconstriction and asthma [40], neutrophil monocyte accumulation in active muscles, and secretion of IL-1, IL-β, and TNF-α [41]. The protective anti-inflammatory effect of vitamin C has been proven in different models of lung and liver injury. In lung fibrosis induced by paraquat (10 mg/kg i.p.) the administration of vitamin C (150 mg/kg/day i.p.) for seven days diminished neutrophil migration by reducing the secretion of IL-17. The authors also reported in the histopathology analysis, anti-fibrotic benefits by less collagen deposition and the pro-fibrotic mediator TGF-β. Together, antioxidant enzymes SOD and catalase were elevated. In the liver, vitamin C has also been reported as having an anti-apoptotic and antioxidant effect at low doses (5 mg/kg), rather than at high doses (500 mg/kg) which may promote oxidation in a paraquat poisoning rat model [42].

It should be noted that, to date, no studies, to our knowledge, have been reported that prove the effectiveness of gas exchange and lung histology modifications with the use of silymarin in physical exercise models. In fact, a small number of studies have been conducted on the use of plants or their active compounds, to evaluate their impact on athletic performance. The study with the *Andiantum capillus-veneris* (Ac-v) extract, a species for which antiapoptotic activity has been described, demonstrated that in treated rats with 500 mL/g/kg body weight (bw)/day for 3 weeks exposed to hypoxic conditions, followed by 6 weeks of an interval training protocol, the expression of p53 protein and of TNF-α was reduced inversely, while the respiratory surface increased. These apoptotic indicators were altered under hypoxic conditions and treatment with the Ac-v extract acted against damaging events [43]. In relation to silymarin and the protective effects on lung tissue, in a model of acute lung damage by aspiration with hydrochloric acid (HCl) (1.2 mL/kg), oral treatment with silymarin at 200 mg/kg for 7 days reduced the indicators of histopathological damage caused in the lung by exposure to HCl. It was also found that silymarin favored the Nrf2/HO-1 pathway, decreasing inflammatory and fibrosis markers as well as anti-apoptotic and cell proliferation parameters [44].

In a study conducted by Zhao et al. [45] in which the authors used 200 mg/kg of silymarin for 3 days in a rodent model, stress induced by malonaldehyde (MDA) was improved. Additionally, an increase was reported in the activity of the antioxidant enzymes superoxide dismutase (SOD), catalase (Cat), and glutathione peroxidase (GPx) in lung tissue and in the serum of rats intoxicated with paraquat at a dose of 30 mg/kg to induce lung injury. These results were related to the induction of the Nrf2, HO-1, and NQO1 pathways. Decrease in histopathological damage and inflammatory mediators, reduction in cellular infiltrate, myeoloperoxidase activity, and the ratio of wet lung weight/dry weight (W/D) were also observed. In conclusion, the researchers assumed that silymarin exerts a protective effect on the lung by activating the Nrf2 pathway. Previously, the use with paraquat to induce lung damage had been described in human lung-adenocarcinoma cell lines A549, where Sm reduced toxicity by inducing the expression of the antioxidant genes *Nrf2*, *NQO1*, and *HO-1* after an exposure of 3 h [46]. It is noteworthy that the nuclear factor E2-related factor 2 (Nrf2) is a key regulator of cellular defense. It coordinates a vast range of genes involved in cytoprotection, antioxidant response, and detoxification, and in addition controls cellular processes such as energy metabolism, autophagy, inflammation, and homeostasis [47,48]. Based on this background, it could therefore be pertinent to suggest that silymarin evidently favors lung morpho-physiology by increasing alveolar size and the bronchial muscle, reducing the number of type 2 pneumocytes and the inflammatory response caused by 8-week of ET, contributing to gas exchange and O_2_ distribution from the blood to the tissues.

To highlight our previous research, we could observe that the pre-training session administration of silymarin diminished muscle and myocardium by reducing cracking and inflammation; in addition, hypertrophy of the quadriceps and gastrocnemius muscles were also detected, probably favored due to a superior vascularization that permitted a more effective gas exchange and the appropriate supply of oxygen and nutrients to the active muscle fibers. In that work, we could also observe that the aerobic endurance capacity improved considerably in all of the groups that performed physical exercise, and it was clear that the training practice led to adaptive physiological changes after 8 weeks of regular exercise training, promoting the delivery and uptake of oxygen in the rodent organism. The administration of silymarin significantly optimized the race time and distance covered when compared with the other groups in the maximal effort test. In turn, it is suggested that silymarin helped to improve physical performance and induced myocardium and quadriceps/gastrocnemius muscle recovery and remodeling and improved body composition, by expanding lean muscle mass, after an 8-week exercise-training program [5]. To continue with this line of research, in the present work the hepatic changes included an increase in vascularization, specifically in the groups treated with antioxidants, and also the reduction of steatosis in all of the trained groups, with an even greater reduction observed in the group with silymarin, which was remarkably similar to that of the control group. Under supervised training, physical exercise appears to effectively reduce the percentage of fat in the liver (4.7%, *p* < 0.05), which was correlated with improvement in respiratory fitness and increased peripheral insulin sensitivity despite not having increased the production of glucose in patients with nonalcoholic hepatic steatosis (NASH) [49]. It was reported that 12 weeks of supervised exercise in men reduced intrahepatic lipid content in persons with and without NASH with normal BMI, highlighting the importance of exercise in liver lipid metabolism [50].

The type of exercise that generates the greatest benefits has indeed been a subject of study, that is, whether aerobic resistance or strength training is the most effective. Aerobic training protocols have been tested in several trials, where a decrease in liver damage indicators, such as the transaminases, alanine aminotransferase (ALT) and aspartate aminotransferase (AST), has been found, as well as a reduction in histological markers of liver injury in NASH [51]. However, recent studies also ensure that resistance training decreases the amount of liver lipids by up to 13% after 8 weeks of strength training [52]. Apparently both types of training exert different mechanisms, by which they reduce the content of liver lipids [53]. Aerobic endurance exercise appears to activate the lipolytic pathway by regulating the protein-1 uncoupling pathway (UCP-1) and PPAR-ƴ, modifying the level of adipokines, while resistance exercise promotes the hypertrophy of type II muscle fibers, altering the level of myokines, activating the glucose transporter 4, calveolins, and AMP-activated protein kinase (AMPK) [54].

The inflammatory reaction, the presence of Kupffer cells, mitosis, and apoptosis/necrosis in the ET group were similar to those of the control group, while the ET+Aa group underwent a low inflammatory reaction and, in the ET+Sm group, this was null. A low level of nuclear hyperchromasia and moderate nuclear enlargement were exhibited. The presence of Kupffer cells in the ascorbic acid and silymarin groups was high. There were no indicators of mitosis, hyperchromasia, or hepatocyte nuclear enlargement. The ET+Aa group demonstrated a low apoptosis/necrosis appearance, while this in the ET+Sm group was null. There is a connection among the indicators of inflammation, hyperchromasia, and nuclear enlargement; such phenomena have been described because of the increase in oxidative stress. The amplified immune reactivity in the liver is regulated by chemical mediators of inflammation and the activation of Kupffer cells, while the increase of nucleus size and coloration could increase with cell division and this increase is stimulated by liver damage or aggression, considering that hepatocytes are extremely sensitive to injury and possess an impressive proliferative capacity. Kupffer cells play an especially important role in the inflammatory response of the liver; their strategic location in the hepatic sinusoids allows them to efficiently engulf pathogens that enter the hepatic circulation. They function as protectors in different situations, such as drug or xenobiotic intoxication; in fact, plasticity in the liver might be the result of the activation state of resident Kupffer cells and/or the recruitment of new monocytes/macrophages in the liver [55]. The functional phenotype of macrophages is associated with the progression of different chronic diseases of the organ, such as NASH, ALD, fibrosis, and hepatocarcinoma [56]. It would be reasonable to assume that the increase in physical training generated increased damage and is therefore one of the observed responses included in the activation of Kupffer cells.

The effect that physical training may have on markers of ischemic reperfusion injury (IRI) in the liver has recently been described. Yazdani et al. [57], employing an IRI model, found that pre-conditioning with physical exercise in mice mitigated the inflammatory response by IRI, protecting the liver from lesions and metastasis. The mice with IRI previously subjected to physical training attenuated necrosis and inflammation markers, such as the formation of neutrophils and neutrophil cell traps (NET); likewise, the expression of liver-cell endothelial adhesion molecules was attenuated. The trained group presented a different population of macrophages, expressing phenotypic *M2* genes. In a metastatic model, training led to the occurrence of less metastasis in mice after 3 weeks of ischemic reperfusion; at the same time, tumor-suppressor T cells were increased in the tumor microenvironment. This research proposes that physical training can serve as a potential non-pharmacological treatment by reducing liver IRI metastases prior to possible surgery.

Despite the few studies conducted to test the ergogenic effect of silymarin on exercise, some evidence has shown changes in body weight in silymarin treatment under certain conditions. Alozy et al. [58] detected a significant increase in body-weight gain (*p* < 0.05) in rats treated with 200 mg/kg/day alone or in combination with 5 mg/kg/day of cyclosporine after 15 days of treatment and continuing for 30 and 45 days. The group exclusively treated with silymarin registered the highest weight gain. Taken together with the reduction of the biochemical parameters of blood glucose, cholesterol, ALT, AST, MDA, and alkaline phosphatase (PHA), and the decrease of the hepatic histopathological indicators, these suggested a protection on the part of silymarin against cyclosporine-induced damage. In our study, it could be observed that silymarin improved lipid content in liver tissue, as well as histological liver damage, and reduced inflammatory response induced by exercise. In agreement with the previous evidence, it is possible that the significant reduction of lipid content in the silymarin group could be due to the inherent effect that silymarin exerts on hepatic lipid metabolism and liver protection, considering that exercise itself improved steatosis in all trained groups, which is supported by the evidence discussed.

Recently, in the study of Docklalova et al. [59], sport horses were feed with milk-thistle seed cakes (400 g/day) for 56 days and exposed to heavy physical exercise (2 h of regular combined driving training). The results showed a general positive effect on health of the horses, but significant differences were found in AST, non-esterified fatty acids (NEFA), cortisol, and phosphorus (Pi). Higher levels of albumin were observed compared to the control, possibly due to a better use of the protein contained in the milk-thistle seed cakes and the proteosynthetic effect attributed to silymarin. The lower NEFA values in horses feed with milk-thistle seed cakes after exercise were associated with a higher utilization of NEFA during physical training. Moreover, a rapid return of cortisol to control levels was reported in experimental group after heavy exercise contributing to recovery. This could reflect the mentioned ergogenic effect of silymarin on physical performance.

## 5. Conclusions

Physical exercise induces acute and chronic responses according to the type, volume, and frequency of the workload. Consequently, the organism engages in the adaptation of the morphophysiology in order to adjust to these changes. In this study, the effect of silymarin supplementation on histological modifications in lung and liver tissue was evaluated. After an 8-week treadmill ET protocol in rats, significant changes were observed in pulmonary architecture, such as an increase of alveolar size, bronchial muscle hypertrophy, and vascularization, suggesting the possible enhancement of gas exchange, better O_2_ delivery, and CO_2_ removal from the tissues, thus promoting a better pulmonary ventilation capacity. This data could be related to the preceding findings regarding the optimization of maximal endurance capacity. In parallel, lipid content in liver was attenuated, importantly, in the group treated with silymarin. Therefore, it is suggested that the administration of silymarin could improve hepatic liver content, the indicators of inflammation in liver tissue, and vascularization in a very similar way to ascorbic acid. The findings in this work are closely related with the ergogenic effect found in our previous work and with the limited evidence found regarding the use of silymarin in athletic performance. In conclusion, the results of this research suggest histological modifications in lung and liver that could be associated with better recovery and performance during physical training. Nevertheless, these results must be accompanied by other biochemical and molecular markers of recovery and damage to support this theory.

## Figures and Tables

**Figure 1 jfmk-06-00072-f001:**
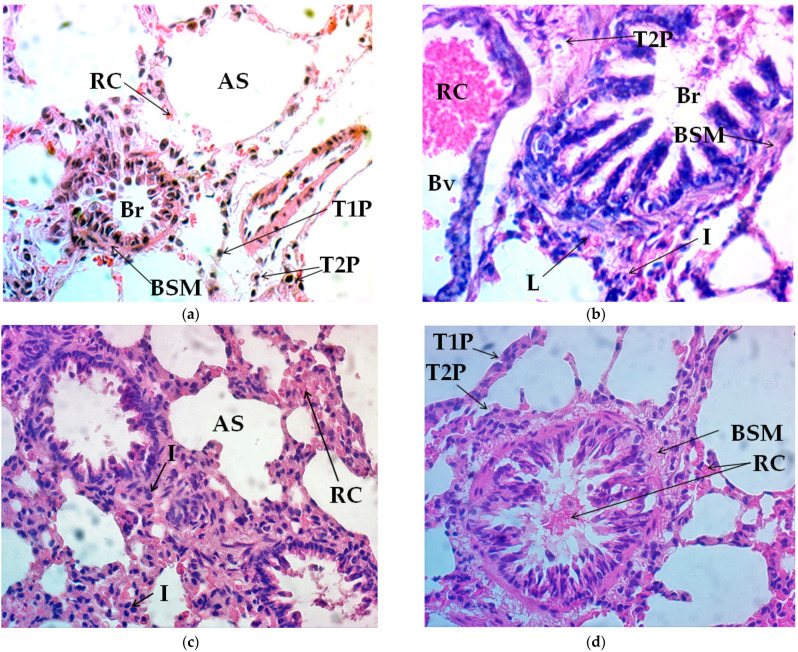
Histological modification of lung tissue in hematoxylin and eosin (H&E) stain (40×). (**a**) control group, (**b**) exercise-training (ET) group, (**c**) exercise-training and ascorbic acid group (ET+Aa), and (**d**) exercise-training and silymarin group (ET+Sm). AS, alveolar space; RC, red cells; T1P, type 1 pneumocytes; T2P, type 2 pneumocytes; Br, bronchiole; BSM, bronchial smooth muscle; Bv, blood vessel; L, lymphocytes; I, inflammation.

**Figure 2 jfmk-06-00072-f002:**
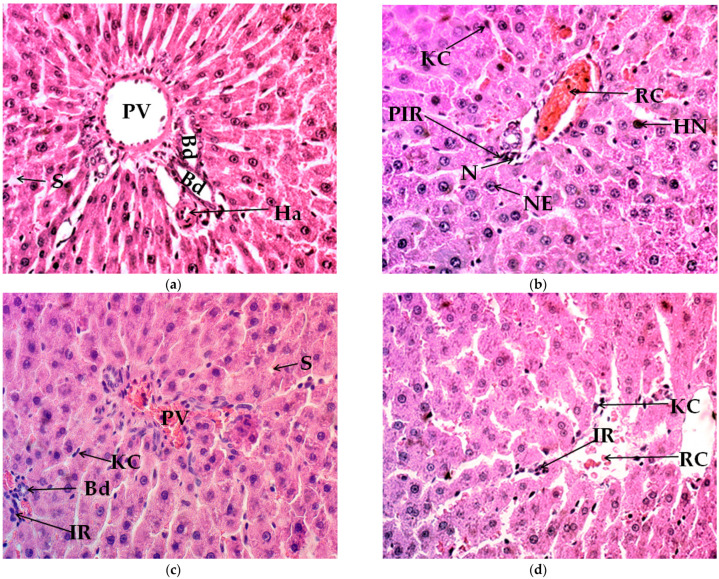
Histological modification of liver tissue in hematoxylin and eosin (H&E) stain (40×). (**a**) control group, (**b**) exercise-training group (ET), (**c**) exercise-training and ascorbic acid group (ET+Aa), and (**d**) exercise-training and silymarin group (ET+Sm). PV, portal vein; Bd, bile duct; S, steatosis; Ha, hepatic arteriole, RC, red cells; KC, Kuppfer cells; PIR, portal inflammatory response; IR, inflammatory response; NE, nuclear enlargement; HN, hyperchromatic nuclei; N, necrosis.

**Table 1 jfmk-06-00072-t001:** Histological changes in lung tissue induced by the 8-week exercise training program in rats supplemented with ascorbic acid and silymarin.

Group	Alveolar Size	Type 1 Pneumocytes	Type 2 Pneumocytes	Vascularization	Inflammation	Bronchial muscle	Connective Tissue	Alveolar Wall Thickness	Pleural Tissue Thickness
Control	+/++	++++	++	++	+	+	+	++	+
ET	+++	++++	+	+++	++/+++	+++	++	++/+++	++
ET+Aa	+++	++++	++	+++	++	+++	+	++	+
ET+Sm	++++	++++	++	++++	+	++++	+	+	+

Criteria: 0 = Null (no presence/appearance), + = Low (minimal presence/appearance), ++ = Moderate (intermediate presence/appearance), +++ = High (important presence/appearance), ++++ = Remarkably high (notable presence/appearance).

**Table 2 jfmk-06-00072-t002:** Histological changes in liver tissue induced by the 8-week exercise training program in rats supplemented with ascorbic acid and silymarin.

Group	Inflammatory Response	Kupffer Cells	Vascularization	Steatosis	Mitosis	Apoptosis/Necrosis	Hyperchromatic Nuclei	Nuclear Enlargement
Control	0/+	++	+	+++	0	0	0/+	0/+
ET	0/+	++	+++	++	0	0	+	++
Et+Aa	+	+++	++++	+	0	+	0	0
Et+Sm	0/+	+++	++++	+	0	0	0	0

Criteria: 0 = Null (no presence/appearance), + = Low (minimal presence/appearance), ++ = Moderate (intermediate presence/appearance), +++ = High (important presence/appearance), ++++ = Remarkably high (notable presence/appearance).

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
