# Peer review of "Effect of Silymarin Supplementation in Lung and Liver Histological Modifications during Exercise Training in a Rodent Model"

_jfmk, 2021, doi:10.3390/jfmk6030072_

Round 1
Reviewer 1 Report
The authors presented data on the effect of silymarin supplementation in histological changes in the lung and liver during exercise in a rodent model. I would like to congratulate the authors for the work, which in my opinion is well written and structured. I have no particular comments to add.
Author Response
Comments and Suggestions for Authors
Point 1. The authors presented data on the effect of silymarin supplementation in histological changes in the lung and liver during exercise in a rodent model. I would like to congratulate the authors for the work, which in my opinion is well written and structured. I have no particular comments to add.
Response.
Thank you for your unvaluable comments, for taking your time to revise this manuscript, we are grateful for response.

Reviewer 2 Report
The authors examined histological changes in the lungs and liver after 8 weeks of treadmill training in male Wistar rats supplemented with ascorbic acid or silymarin. The topic is new, but the manuscript has major flaws in reporting and linking variables together. In addition, to examine the silymarin supplement, the study needed a control groups plus silymarin which is unfortunately missed. Overall, the manuscript is poorly written and needs to be largely reviewd.
Introduction
The introduction is vague and does not provide an adequate rationale for the study. The first paragraph is very general and does not relate to the topic of the study. In the second paragraph, I would have expected the authors to explain the literature with more detail. Although the opic of study refers to the lungs and liver structure, there is no functional examination (gas exchange or metabolic tests), so all arguments on O2 and CO2 exchange and glycogen levels are considered as speculations or unrelated. Here the authors could explain what will be the advantage of the histological examination and why the expected changes after 8 weeks of aerobic training.
Methods
I think the ascorbic acid group can be considered as the main group rather than control groups and can also be added to the title and subsequently all other parts.
Results
One of the most difficult parts of the current study is the interpretation of the results. Since the changes in the training groups are very similar, it is very difficult to find the superior groups, and this should also be taken into account in the discussion.
Discussion and conclusions
The confusing part of the study is the discussion section. The discussion part is very long and mostly unrelated to the results of the study. Here I suggest to remove all unnecessary explanations and mechanisms and focus only on the results of the current study and compare it with other similar studies shortly. Also, since it is not clear whether the difference between groups is significant or not, the conclusion must include all training groups.

Author Response
Comments and Suggestions for Authors
Point 1. The authors examined histological changes in the lungs and liver after 8 weeks of treadmill training in male Wistar rats supplemented with ascorbic acid or silymarin. The topic is new, but the manuscript has major flaws in reporting and linking variables together. In addition, to examine the silymarin supplement, the study needed a control groups plus silymarin which is unfortunately missed. Overall, the manuscript is poorly written and needs to be largely reviewed.
Response 1.Thank you for your comments, for reasons beyond our scope of space and budget, it was not possible to include one more study group, which is why it was not considered to have another control group plus silymarin in this research. However, it is being considered for future research in our study group.
Point 2. Introduction
The introduction is vague and does not provide an adequate rationale for the study. The first paragraph is very general and does not relate to the topic of the study. In the second paragraph, I would have expected the authors to explain the literature with more detail. Although the topic of study refers to the lungs and liver structure, there is no functional examination (gas exchange or metabolic tests), so all arguments on O2 and CO2 exchange and glycogen levels are considered as speculations or unrelated. Here the authors could explain what will be the advantage of the histological examination and why the expected changes after 8 weeks of aerobic training.
Response 2. In this part we consider important to explain the functional changes in lung and liver, besides this study did not include functional examination as you certainly mention, because the intention is to give a general overview of the importance to the connection between
As you suggest, the importance of lung and liver histological analysis is explained you can find on lines 81 to 92 as well as the corresponding references in yellow color. Regard to liver, until now the available information of tissue changes during physical exercise is very limited, thought we consider important to report what we could find in this study.
Point 3. Methods
I think the ascorbic acid group can be considered as the main group rather than control groups and can also be added to the title and subsequently all other parts.
Response 3. Thank you for the suggestion. The evidence regarding the use of silymarin as ergogenic in sports is null. We decided to use ascorbic acid as a reference because possess scientific evidence as antioxidant in sports training protocols. Even, the International Society of Sports Nutrition at the document regard to “ISSN exercise & sports nutrition review update: research & recommendations” published on 2018, stated vitamin C in category I according to the classification of dietary supplements in sports, considering as follows:
- Strong Evidence to Support Efficacy and Apparently Safe: Supplements that have a sound theoretical rationale with the majority of available research in relevant populations using appropriate dosing regimens demonstrating both its efficacy and safety.
On the other hand, International Olympic Committee on its “IOC consensus statement: dietary supplements and the high-performance athlete” published in 2018, point out that vitamin C can be consider as a nutrient that may quenches reactive oxygen species reducing oxidative stress and augmenting immunity. Evidence suggest that reduces interleukin-6 and cortisol responses to exercise in humans. Besides, consider vitamin C as part of the supplements that may assist with training capacity, recovery, muscle soreness and injury management.
That is why we consider trying with vitamin C in this study, however, in this research, we attempt to show if silymarin has a similar or higher effect than ascorbic acid to reduce inflammation or improve some indicators of recovery in the lung and liver after exercise bouts.
This additional information is found on lines 337-360.
References:
- Kerksick CM, Wilborn CD, Roberts MD, Smith-Ryan A, Kleiner SM, Jäger R, Collins R, Cooke M, Davis JN, Galvan E, Greenwood M, Lowery LM, Wildman R, Antonio J, Kreider RB. ISSN exercise & sports nutrition review update: research & recommendations. J Int Soc Sports Nutr. 2018 Aug 1;15(1):38. doi: 10.1186/s12970-018-0242-y. PMID: 30068354; PMCID: PMC6090881.
- Maughan RJ, Burke LM, Dvorak J, Larson-Meyer DE, Peeling P, Phillips SM, Rawson ES, Walsh NP, Garthe I, Geyer H, Meeusen R, van Loon LJC, Shirreffs SM, Spriet LL, Stuart M, Vernec A, Currell K, Ali VM, Budgett RG, Ljungqvist A, Mountjoy M, Pitsiladis YP, Soligard T, Erdener U, Engebretsen L. IOC consensus statement: dietary supplements and the high-performance athlete. Br J Sports Med. 2018 Apr;52(7):439-455. doi: 10.1136/bjsports-2018-099027. Epub 2018 Mar 14. PMID: 29540367; PMCID: PMC5867441.
Point 4. Results
One of the most difficult parts of the current study is the interpretation of the results. Since the changes in the training groups are very similar, it is very difficult to find the superior groups, and this should also be taken into account in the discussion.
Response 4. Histology was analyzed again after your suggestion. An imagen analysis was performed to detect additional changes to what was reported previously. The quality of the figures was corrected to show with more clarity changes in tissues. The changes are visible on Figures 1c and 2c.
Point 5. Discussion and conclusions
The confusing part of the study is the discussion section. The discussion part is very long and mostly unrelated to the results of the study. Here I suggest to remove all unnecessary explanations and mechanisms and focus only on the results of the current study and compare it with other similar studies shortly. Also, since it is not clear whether the difference between groups is significant or not, the conclusion must include all training groups.
Response 5. Thank you for the comments in this part. As you suggest, it was removed part of explanation of the mechanism, but we consider necessary to keep some of them in order to connect the results.
It was difficult to find other similar studies in the context of basic research of antioxidants and tissue modifications and performance. However, use the available studies concerning to silymarin effect in physical performance and in lung and liver injury models. Additional references were adjusted in the manuscript using EndNote X9 program.

Round 2
Reviewer 2 Report
I would like to congratulate the authors on their work